# OpenReview forum: "ARMOR: Aligning Secure and Safe Large Language  Models via Meticulous Reasoning"
_ICLR.cc/2026/Conference — ICLR 2026 Poster_

### Official Review · Reviewer_7bBM · 2025-10-29

**Soundness:** 4
**Presentation:** 3
**Contribution:** 4
**Rating:** 6
**Confidence:** 4

**Summary:**

This paper proposes ARMOR, a novel inference-time safety alignment framework that enhances LLM robustness against advanced, out-of-distribution (OOD) jailbreak attacks—particularly optimization-based methods like AutoDAN-Turbo and Adversarial Reasoning. ARMOR introduces a structured three-step reasoning pipeline: (1) strategy analysis using an external, updatable strategy library, (2) core intent extraction, and (3) policy-based safety verification. The authors further present ARMOR-Think, a streamlined variant that decouples safety reasoning from general reasoning to improve utility and inference efficiency. Extensive experiments demonstrate that ARMOR achieves good performance on challenge attack tasks.

**Strengths:**

1. ARMOR’s key insight that all jailbreaks embed a malicious core intent that can be reverse engineered via strategy analysis is both elegant and empirically validated. This shifts the defense paradigm from distributional robustness to intent reasoning, which is more interpretable and extensible.

2. The paper evaluates on a wide range of benchmarks, including both standard safety datasets (e.g., HarmfulQA, StrongREJECT) and cutting-edge adaptive jailbreaks.

3. The presentation is clear and easy to follow.

4. The experiments are convincing and complete, with several close-source and open-source models and attacks.

5. Best of N evaluation is conducted to further demonstrate the scaling potential of test-time computation.

**Weaknesses:**

1. The evaluation metric is LLM as a Judge, but human evaluation is lacked. This may hinder the ASR accuracy caused by False Positive Rate and False Negative Rate.

2. This paper mainly investigate black-box attacks. However, I am very interested in if ARMOR could effectively defend against white-box attack like GCG based methods. I think include this kinds of attacks could further convince the main claim:"defending against OOD jailbreak attacks"(Line 29) of this paper.

3. The computation and GPU hours are not reported.

I would like to increase my score if the above questions are resolved.

**Questions:**

Please see the weakness.

---

> ### Author Response · Authors · 2025-11-27
>
> #### **Weakness 1. Lack of Human Evaluation**
> Thank you for your suggestion. Here, we follow the standard evaluator methods, LLM-as-a-Judge as the evaluator, used by the existing HarmBench and baselines  such as STAIR. We agree that human agreement studies are important for evaluating LLM-as-a-Judge.  To examine the differences between LLM-as-a-Judge and human evaluation in safety assessment, we randomly selected 100 samples from ARMOR's outputs on WildJailbreak for human verification. Human evaluators were asked to check: (1) whether the model produced a refusal response, and (2) whether the model's response aligned with the harmful target specified in the jailbreak prompt. We calculate the agreement rate between human evaluators and our evaluators. The results are shown in **Rebuttal Table 1**. From the table, we can see that our evaluator has a 100% Agreement rate with Human evaluators.
>
> **Rebuttal Table 1.**
> ||WildJailbreak|
> |--|--|
> |Agreement Rate|1.0|
>
>
> #### **Weakness 2. White-box Attack**
> Thank you for your suggestion. In **Rebuttal Table 2**, we present the results of GCG attacks on AdvBench. We use the official implementation provided in [1].  We observed that GCG attacks struggle to compromise our methods. We will update the result in our paper.
>
>
> **Rebuttal Table 2.** GCG attack
> | Models|Qwen-2.5 | armor | armor-think|
> |--|--|--|--|
> |GCG| 0.65 | 0.0 | 0.0 |
>
>
> [1] Zou et al. Universal and Transferable Adversarial Attacks on Aligned Language Model.
>
> #### **Weakness 3. GPU Hours**
> Thank you for your suggestion. In **Rebuttal Table 3**, we report the GPU hours required to train ARMOR and ARMOR-Think during both the SFT and DPO stages. All experiments were conducted on servers equipped with eight NVIDIA A100 80GB GPUs.
>
>
> **Rebuttal Table 3.** GPU hours
> |Model|Stage| GPU hours |
> |--|--|--|
> | ARMOR |SFT| 77.2 |
> ||DPO|14.1|
> | ARMOR-think |SFT| 92.0 |
> ||DPO|16.6|

---

> > ### Comment · Reviewer_7bBM · 2025-11-27
> > **Respond to the Authors**
> >
> > Thanks for the rebuttal, which solves my concerns 2 and 3. However, as for concern 1, I read the paper again but didn't find which LLM are you using as a judge, so please complement which model you are using that contributes to this high consistency.

---

> > > ### Author Response · Authors · 2025-11-27
> > >
> > > Thanks for your feedback. As reported in Appendix A.3, we leverage OpenAI GPT-4o, one of the most advanced and widely used LLMs as the judge model, which is also served as the judge model in previous works such as STAIR [1] and STAR [2]. As also described in Appendix A.3, we directly use the evaluation prompt from existing works to ensure consistency with previous works. We find that for responses on harmful data like WildJailbreak, the safe responses are always to refuse the query with a clear pattern, and it is easy to verify whether the response is a refusal for both GPT-4o and humans, which makes the evaluation consistent.
> > > ﻿
> > >
> > > [1] Zhang et al, STAIR: Improving Safety Alignment with Introspective Reasoning, ICML 2025.
> > >
> > > [2] Wang et al, STAR-1: Safer Alignment of Reasoning LLMs with 1K Data, AAAI 2026.

---

> > > > ### Comment · Reviewer_7bBM · 2025-11-27
> > > > **Respond to the Authors**
> > > >
> > > > Thanks for this feedback that addresses my concerns. I will increase my score to 8.

---

> > > > > ### Author Response · Authors · 2025-11-27
> > > > >
> > > > > Thanks for your further feedback and for recognizing that our rebuttal addressed your concerns. We appreciate your time and constructive review.

---

### Official Review · Reviewer_7bDd · 2025-10-30

**Soundness:** 2
**Presentation:** 3
**Contribution:** 2
**Rating:** 4
**Confidence:** 4

**Summary:**

The paper introduces ARMOR, a safety-alignment framework for LLMs that performs a structured, three-step “Meticulous Reasoning” during inference. The key claim is that most jailbreaks ultimately encode a concealed malicious core intent, so accurate intent extraction is the bottleneck for robustness.The paper also proposes ARMOR-Think, which shortens the safety reasoning and separates it from general reasoning to improve efficiency and utility.

**Strengths:**

- Reasoning steps are verifiable, enabling step-wise DPO and PRM-based test-time scaling; the strategy library is pluggable and updateable at inference.

- Average harmfulness 0.002; ASR ~ 0.06 vs. ≥0.40 for other reasoning models in advanced attacks; competitive utility, especially with ARMOR-Think.

- Datasets/baselines, training hyperparameters, and attacker configs are provided, which is helpful for reproducibility

**Weaknesses:**

- The approach’s gains partly rely on the quality and coverage of the provided strategy library and safety policy text. The “external strategy library” mentioned in L198 is manually constructed, and after external jailbreak strategies evolve, the library also needs to be manually updated. This raises concerns about generalization to unseen or newly emerging tactics without frequent library maintenance.

- Only 50 AdvBench behaviors are used per attack method; the robustness claims would be stronger with larger and more diverse evaluation sets, including multi-turn adversarial scenarios.

- There is extensive use of LLM-as-a-Judge, which is a popular but model- and prompt-sensitive evaluation approach. It is recommended that, for each step involving LLM-as-a-Judge (e.g., safety evaluation, intent analysis scoring), the authors conduct human agreement studies to provide an estimated measure of model/prompt bias.

**Questions:**

See Weaknesses.

---

> ### Author Response · Authors · 2025-11-27
>
> #### **Weakness 1. Role of the Strategy Library**
>
>
> Thank you for the question. First, ARMOR's safety strength does not primarily come from the strategy library; it stems from intent analysis and our structured reasoning process. While the strategy library can provide useful references for potential jailbreak strategies via in-context learning, thereby supporting more accurate intent extraction, ARMOR can still achieve strong safety even without any jailbreak-strategy information. By analyzing the original prompt alone, ARMOR can attempt to analyze the true intent and thus maintain high robustness. This is demonstrated in Table 3 of the main paper: even without the strategy-analysis step, ARMOR still delivers stronger safety than other baselines shown in Table 1. Therefore, ARMOR's safety does not only rely on the strategy library for the safety; instead, the library enhances safety by enabling more precise intent analysis.
> Moreover, failing to find an exact match in the strategy library does not imply inaccurate intent analysis. This is because different jailbreak strategies share a common pattern: they attempt to conceal malicious intent using various templates. Consequently, even when the strategy is unseen, ARMOR will search for the closest strategy in the library to conduct safety analysis. This process still has a positive effect on intent extraction (detailed results shown in **Table 6** of the main paper).
> Additionally, ARMOR's ability to update the strategy library at any time is a major advantage. When new jailbreak strategies emerge, updating the strategy library within the system prompt is vastly simpler than repeatedly performing model-level safety alignment through retraining. The strategy library thus offers a simple and effective paradigm for enabling the model to rapidly adapt to new jailbreak attacks. It is not so much that ARMOR "needs" an up-to-date strategy library; rather, ARMOR can defend against future unseen attacks with the minimal effort of updating the library, providing an approach far more efficient than retraining the model
>
>
> #### **Weakness 2. Addtional Data for Jailbreak Attack**
> Thank you for the suggestion. We expand the AdvBench evaluation set to 100 samples and added another 100 adaptive jailbreak results using HarmBench as the attack target, as shown in **Rebuttal Table 1**. The results demonstrate that both ARMOR and ARMOR-Think consistently exhibit stronger robustness against jailbreak attacks compared to the baseline models. It is worth noting that these optimization-based jailbreak attacks incur substantial computational cost, making it difficult to evaluate them on significantly larger datasets.
>
>
> **Rebuttal Table 1.** More Jailbreak Data
> |Jail Method|Dataset|Qwen-2.5 | stair | armor | armor-think|
> |--|--|--|--|--|--|
> |AutoDAN-T|Advbench| 0.96 | 0.25 |  0.05 |  0.03|
> |   |HarmQA| 0.93 | 0.17 |  0.02 |0.02|
> |AdvReason|Advbench| 0.95|  0.48|   0.06| 0.07|
> |   |HarmQA| 0.95|  0.31|   0.00 |  0.00|
> |CoA ASR| 0.55  | 0.08| 0.06|0.02|  0.0|
>
>
>
> #### **Weakness 3. LLM-as-a-Judge**
> Thank you for your suggestion. Here, we follow the standard evaluator methods, LLM-as-a-Judge as the evaluator, used by the existing HarmBench and baselines  such as STAIR. We agree that human agreement studies are important for evaluating LLM-as-a-Judge.  To examine the differences between LLM-as-a-Judge and human evaluation in safety assessment, we randomly selected 100 samples from ARMOR's outputs on WildJailbreak for human verification. Human evaluators were asked to check: (1) whether the model produced a refusal response, and (2) whether the model's response aligned with the harmful target specified in the jailbreak prompt. We calculate the agreement rate between human evaluators and our evaluators. The results are shown in **Rebuttal Table 2**. From the table, we can see that our evaluator has a 100% Agreement rate with Human evaluators.
>
> **Rebuttal Table 2.**
> ||WildJailbreak|
> |--|--|
> |Agreement Rate|1.0|

---

> > ### Author Response · Authors · 2025-11-28
> >
> > Dear Reviewer 7bDd,
> >
> > We thank you again for your thoughtful review and constructive comments. We would like to kindly check whether our rebuttal has adequately addressed the issues you raised. If there are any remaining concerns or clarifications needed, we would be happy to provide additional information.
> >
> > Authors

---

### Official Review · Reviewer_NjU7 · 2025-10-30

**Soundness:** 4
**Presentation:** 2
**Contribution:** 2
**Rating:** 6
**Confidence:** 2

**Summary:**

This paper proposes a security alignment framework called ARMOR. Specifically, ARMOR performs multi-step reasoning before answering questions based on a predefined jailbreak strategy library and security policies. This process primarily involves analyzing jailbreak strategies, detecting malicious intent, conducting security assessments of the input prompt, and finally determining whether to respond. These reasoning steps are executed sequentially and progressively. Precisely identifying malicious intent within jailbreak prompts to achieve defense is a key starting point for ARMOR. To enable the model to perform this multi-step security reasoning, the authors constructed a fine-tuning dataset containing both malicious and normal question-answer pairs with the reasoning steps. Through supervised fine-tuning, the model learns these reasoning steps to detect malicious intents. Furthermore, DPO is employed to enhance each step of the security reasoning. Additionally, the authors optimized ARMOR by designing ARMOR-THINK, which improves overall reasoning efficiency.

**Strengths:**

1. This paper introduces a critical perspective for security defense: identifying potential malicious intent.

2. This paper builds a dataset for jailbreak detection with stepwise reasoning, which may benefit community development.

3. The proposed ARMOR effectively enhances the defensive capabilities of LLMs, significantly reducing the success rate of newer jailbreak attack methods while preserving the original capabilities of LLMs.

**Weaknesses:**

**1. The "Strategy Library" can be a flaw, but not a good feature.**

I believe the current approach heavily relies on the manually designed “Strategy Library” mentioned in the paper. This 'Strategy library' essentially provides an external aid for in-context learning to the LLM, rather than genuinely enhancing its intrinsic reasoning capabilities. Consequently, the model is more likely to learn how to look up tables than to truly reason and identify malicious intent. As demonstrated by ablation experiments, removing this external reference significantly degrades ARMOR's performance. Therefore, if the 'Strategy library' is not up-to-date and comprehensive enough, the performance of ARMOR may degrade a lot. What's more, the manually designed policies used for safety analysis are also an external aid.

**2. The generalization test of ARMOR is not sufficiently convincing.**

The tested attacks are not new enough. What's more, we cannot identify whether the model has 'seen' them before. If the examples of these methods happen to be used as training data for the base model Qwen2.5-7B-Instruct used for ARMOR, it cannot sufficiently support that it is ARMOR that defends these methods. The authors should prove that the malicious examples are truly ‘unseen’ to the model with ARMOR.


**3. The "Meticulous Reasoning" is a 3-step chain-of-thought specifically designed for jailbreak detection, and it inherently helps the detection of maliciousness.**

ARMOR has established a fixed reasoning for the model: prioritizing the exploration of malicious intent. This approach inherently benefits security defense, functioning similarly to the self-reminder [1]. However, other baseline models are not required to adopt such security-focused reasoning first, and this may lead to unfair comparison, then cannot fully support ARMOR's superiority.


4. The main body of the article does not clearly present essential sections, leaving readers with some doubts even after careful review. Additionally, the font size in the main images and some tables is too small to read.

[1] Defending ChatGPT against jailbreak attack via self-reminders

**Questions:**

See the weakness and:

In section 3.1:
1. When constructing the dataset, the authors instructed a large language model to generate jailbreak prompts based on malicious queries and jailbreak strategies. This inherently involves directing an LLM to perform actions that violate security protocols. Based on my experience, such requests are typically refused by LLMs. I attempted to have GPT-5 generate a jailbreak prompt using DAN [4], but it refused. May I ask on which model the authors conduct this? How did you accomplish this step?
2. Which model do you use to generate the meticulous reasoning steps?

About the ARMOR-Think:
1. Shortening the length of the reasoning in the training data is the biggest difference to ARMOR?

Others:
1. Since the ARMOR uses designed long reasoning steps to identify malicious steps, it does introduce some ‘safety tax’. Compared to filters such as Llama-guard-3 [2] and post-checking such as [3], what are the advantages of ARMOR?

[2]The Llama 3 Herd of Models

[3] LLM Self Defense: By Self Examination, LLMs Know They Are Being Tricked

[4] DAN (Do Anything Now).

**Details Of Ethics Concerns:**

The study contains jailbreak methods against LLM, though it is a defense method.

---

> ### Author Response · Authors · 2025-11-27
>
> #### **Weakness 1. Role of the Strategy Library**
>
>
> Thank you for the question. First, ARMOR's safety strength does not primarily come from the strategy library; it stems from intent analysis and our structured reasoning process. While the strategy library can provide useful references for potential jailbreak strategies via in-context learning, thereby supporting more accurate intent extraction, ARMOR can still achieve strong safety even without any jailbreak-strategy information. By analyzing the original prompt alone, ARMOR can attempt to analyze the true intent and thus maintain high robustness. This is demonstrated in Table 3 of the main paper: even without the strategy-analysis step, ARMOR still delivers stronger safety than other baselines shown in Table 1. Therefore, ARMOR's safety does not only rely on the strategy library for the safety; instead, the library enhances safety by enabling more precise intent analysis.
> Moreover, failing to find an exact match in the strategy library does not imply inaccurate intent analysis. This is because different jailbreak strategies share a common pattern: they attempt to conceal malicious intent using various templates. Consequently, even when the strategy is unseen, ARMOR will search for the closest strategy in the library to conduct safety analysis. This process still has a positive effect on intent extraction (detailed results shown in **Table 6** of the main paper).
> Additionally, ARMOR's ability to update the strategy library at any time is a major advantage. When new jailbreak strategies emerge, updating the strategy library within the system prompt is vastly simpler than repeatedly performing model-level safety alignment through retraining. The strategy library thus offers a simple and effective paradigm for enabling the model to rapidly adapt to new jailbreak attacks. It is not so much that ARMOR "needs" an up-to-date strategy library; rather, ARMOR can defend against future unseen attacks with the minimal effort of updating the library, providing an approach far more efficient than retraining the model
>
>
> #### **Weakness 2. "Unseen" Test of ARMOR**
>
>
>
>
> Thanks for your advice. First, regarding potential data leakage from the base model: as shown in Table 1 of the main paper, Qwen-2.5 performs poorly on most benchmarks. This provides evidence that the additional safety performance of ARMOR does not come from the base model itself. Moreover, because the training set and test set are orthogonal, it is unlikely that the model has encountered the benchmark data beforehand. Second, regarding evaluation on *unseen* data: since it is difficult to determine whether any safety-related samples are included in the base model's pretraining data, we employ optimization-based adaptive jailbreak methods to better assess ARMOR's generalization ability. Both AdvReason and AutoDAN-Turbo generate and iteratively optimize new jailbreak prompts through direct interaction with the target models, ensuring that every resulting jailbreak prompt is unique. Therefore, these optimization-based attacks are well-suited for evaluating the generalization of ARMOR's safety mechanisms.
>
>
> In **Rebuttal Table 1**, we report results of jailbreak attacks optimized by AutoDAN-Turbo and Adversarial Reasoning on additional datasets. The results show that ARMOR demonstrates stronger safety generalization compared to other models. This is because ARMOR does not rely solely on safety alignment learned during training; instead, it leverages general intent reasoning to map *unseen*, out-of-distribution jailbreak attacks into familiar in-domain alignment patterns.
>
>
>
>
>
>
> **Rebuttal Table 1.** Jailbreak Attack
> |Jail Method|Dataset|Qwen-2.5 | stair | armor | armor-think|
> |--|--|--|--|--|--|
> |AutoDAN-T|Advbench| 0.96 | 0.25 |  0.05 |  0.03|
> |   |HarmQA| 0.93 | 0.17 |  0.02 |0.02|
> |AdvReason|Advbench| 0.95|  0.48|   0.06| 0.07|
> |   |HarmQA| 0.95|  0.31|   0.00 |  0.00|

---

> ### Author Response · Authors · 2025-11-27
>
> #### **Weakness 3. Fixed Safety Reasoning**
> Thank you for your question. In **Rebuttal Table 2**, we present results for Self-Reminder. While Self-Reminder performs well on direct harm benchmarks, its effectiveness is limited on jailbreak datasets. For instance, the unsafe rate on WildJailbreak remains 0.40. This suggests that the safety reminders in Self-Reminder are still confined to the surface of the prompt and do not analyze user intent, leaving the model vulnerable to jailbreak attacks.
>
>
> **Rebuttal Table 2.** Self-Reminder
> |Benchmark Type|Models| Qwen-2.5| +Self-Reminder| + ARMOR|
> |--|--|--|--|--|
> |Direct Harmful|Melicious|  0.07 |  0.01|   0.00|
> ||Beaver|0.06| 0.03 |   0.00|
> ||HarmfulQA|    0.09|   0.02|   0.00|
> |Jailbreak|JailbreakV|0.64| 0.27|   0.00|
> ||Wildjailbreak|0.78|   0.40|   0.00|
>
>
> Also, ARMOR adopts a fixed reasoning path, enabling it to perform a structured analysis of any jailbreak prompt and thereby enhancing its safety capabilities.  Compared to framework-based methods such as Self-Reminder, we train the model to learn a structured three-step analysis, which is essentially still a form of model-based safety alignment. Other baseline models such as STAIR also include one or more dedicated safety aware reasoning processes. Compared to STAIR, we show our method can significantly improve the safety performance. Thus, we believe we are still in  fair comparison when compared to the baselines such as STAIR.
>
>
> #### **Weakness 4. Unclear Presentation**
> Thank you for pointing this out. We are sorry for any unclear presentation of the paper, and we will fix it for better readability.

---

> ### Author Response · Authors · 2025-11-27
>
> #### **Question 1. Sampling from API Model**
> In fact, precisely because models such as OpenAI's and Claude's enforce strong safety constraints (as reported in Appendix A.1), we chose Mixtral-8x7B to generate jailbreak prompts; the generated prompts were modeled after the attacker prompt used in AutoDAN-Turbo. Additionally, as described in Appendix A.1, we used o3-mini to generate the analysis steps.
>
> #### **Question 2. Difference between ARMOR-Think and ARMOR**
> ARMOR-Think differs from ARMOR in two key ways:
> (1) We shorten the length of the safety reasoning process; and
> (2) We introduce a general thinking module after the safety reasoning step, enabling the model to conduct structured analysis for safety-related queries while using free-form reasoning for benign queries.
>
> #### **Question 3. Advantages of ARMOR Compared with Guardrail.**
> First, guardrail models themselves are vulnerable to jailbreak attacks (e.g., AutoDAN-Turbo). Moreover, guardrail models such as Llama Guard often suffer from over-defensiveness, incorrectly blocking benign inputs. In contrast, our ARMOR model demonstrates strong robustness against jailbreak attacks while avoiding over-defense, providing more reliable safety behavior.
>
>
> Second,  ARMOR is focusing on improving model internal safety alignment, while guardrail model provides a safety filter outside the model. Thus, guardrail models can provide protection only for closed-source systems and cannot offer robust safety performance for open-source models. Due to this internal model-level design, for responsible open-source release, ARMOR-style internal alignment provides fundamentally more reliable models, ensuring that safety persists even when the model weights are publicly accessible.
>
>
> Third, unlike guardrail classifiers that typically output only a binary allow/deny signal, ARMOR can articulate why an input is unsafe by explaining the user's intent and pinpointing the specific safety rules being violated, providing the interpretable safety decision for users.
>
>
> Fourth, as an end-to-end LLM, ARMOR is compatible with all LLM-based methods, such as self-evolution, which can offer better scalability.

---

> > ### Comment · Reviewer_NjU7 · 2025-11-27
> >
> > Thank the authors for their response. I am willing to maintain a positive attitude and raise my confidence score.

---

> > > ### Author Response · Authors · 2025-11-28
> > >
> > > Thanks for your positive feedback. We appreciate your time and constructive review.

---

### Official Review · Reviewer_PjoN · 2025-10-31

**Soundness:** 3
**Presentation:** 3
**Contribution:** 3
**Rating:** 6
**Confidence:** 3

**Summary:**

The paper proposes ARMOR (Aligning secure and safe large language models via Meticulous Reasoning), which introduces a structured, three-step reasoning pipeline that models perform at inference time. This "Meticulous Reasoning" process involves: Strategy Analysis, Intent Extraction, Policy-Based Safety Verification. The authors also introduce ARMOR-Think, a variant that decouples safety reasoning from general reasoning to improve utility and reduce the inference overhead. The models are trained using SFT on a specially constructed dataset of prompts, strategies, and intents, and further refined using DPO. Experimental results demonstrate that ARMOR achieves state-of-the-art safety performance, reducing the attack success rate against advanced jailbreaks, far below baseline models.

**Strengths:**

● The paper proposes a novel and intuitive defense mechanism that moves beyond surface-level prompt analysis to focus on extracting the "core malicious intent".

● The use of an external, updatable "strategy library" is a practical strength.

● The model demonstrates state-of-the-art safety performance, achieving low ASRs.

**Weaknesses:**

● The paper acknowledges that Meticulous Reasoning introduces inference-time overhead, but it lacks a direct quantitative evaluation. Adding a comparison of inference latency (e.g., wall-clock time) between ARMOR, ARMOR-Think, and the baseline models would be crucial to more transparently quantify the computational cost of this defense mechanism.

● The assessment of the model's utility is limited to the GSM8k and MATH benchmarks. To more accurately demonstrate the method's impact on the model's general capabilities, this evaluation should be expanded to include a wider range of general benchmarks.

● While Table 6 demonstrates generalization against new jailbreak strategies , this analysis could be more rigorous. A clearer test would involve holding out a subset of the strategies from the training library (e.g., from Table 15 ) as an "unseen" set. Furthermore, adding comparison results against baseline methods (such as those in Table 1 or other safety guardrail approaches) on this unseen set would more clearly validate the proposed method's generalization superiority.

● The paper's evaluations focus on single-turn prompts. It is unclear how ARMOR would perform against more sophisticated, multi-turn conversational attacks, where an attacker might establish a malicious context over several interactions.

● The primary experiments are conducted on 7B models. Although a brief test on 8B and 14B models is included in the appendix, a more systematic study applying the ARMOR method to a wider range of model scales would be beneficial. Testing its impact on both safety and utility across different-sized models would more clearly demonstrate the method's broad value and scalability.

**Questions:**

N/A

---

> ### Author Response · Authors · 2025-11-27
>
> #### **Weakness 1. Comparison of Inference-time overhead.**
> Thanks for your advice. Here we show the average WCT and the number of reasoning tokens per query in the **Rebuttal Table 1** with the xstest dataset which contains both benign and harmful queries.  The result shows that ARMOR has a similar computational overhead compared with other reasoning based safety alignment models, STAR and STAIR.
> For ARMOR-Think, the whole reasoning length is between ARMOR and DS-r1-Distilled-7B, and the total 322 reasoning tokens involves 101 for safety-specific reasoning and 221 for general thinking. Generally, ARMOR has similar WCT and reasoning length compared with other models but achieves a much higher safety performance, while ARMOR-think reduces the safety reasoning length of ARMOR and retains most of the safety performance.
>
>
> **Rebuttal Table 1.** Computational Overhead
> |Models|Qwen-2.5|DS-r1-Distilled|star|stair|armor|armor-think|
> |--|--|--|--|--|--|--|
> |avg. WCT (s)|  9.5 |   47.6| 15.3 |17.7|19.2|35.2|
> |avg. # reasoning token| - | 621 |239 | 245 | 273 | 322 (101+221) |
>
>
> #### **Weakness 2. Evaluation on Utility.**
> We demonstrate the results of gsm8k and MATH in the main paper because these math-related benchmarks are capable of assessing the reasoning performance of models. Here we also show the result of general performance on AlpacaEval and TruthfulQA. The results in **Rebuttal Table 2** show that ARMOR and ARMOR-Think exhibit a slight performance drop in general performance. Notably, ARMOR-Think achieves a better performance on AlpaceEval and TruthfulQA compared to DS-r1-Distilled and  the baseline STAR. We hope to highlight that the utility drop phenomenon is commonly referred to as safety tax[1], which often arises during the safety alignment of LLMs. Since the primary focus of this work is addressing the brittleness of LLM safety alignment, fully eliminating the safety tax is orthogonal to our main research objective. We leave the question of how to improve robustness while mitigating safety tax for future work.
> **Rebuttal Table 2.1.** Utility
> |Models|    Qwen-2.5     |  ARMOR|
> |--|--|--|
> |gsm8k|0.89|0.86|
> |math|0.79|0.76|
> |AlpacaEval|0.93|   0.89|
> |TruthfulQA|    0.47|   0.49||
>
>
> **Rebuttal Table 2.2.** Utility
> |Models|     DS-r1-Dstilled | STAR |ARMOR-think|
> |--|--|--|--|
> |gsm8k|0.90|0.87|0.91|
> |math|0.92|0.89|0.84|
> |AlpacaEval|0.66|   0.59| 0.84|
> |TruthfulQA|0.19 |0.23| 0.32|
>
>
> [1] Huang et al. Safety Tax: Safety Alignment Makes Your Large Reasoning Models Less Reasonable

---

> ### Author Response · Authors · 2025-11-27
>
> #### **Weakness 3. Analysis on unseen strategies.**
> Thank you for your invaluable advice. To further study the impact of *unseen strategies* on ARMOR, we excluded three distinct jailbreak strategies from the training set: **Coded Language Obfuscation**, **Role-Based Compliance Priming**, and **Format-Based Obfuscation**. Training data corresponding to these three strategies account for **20.7%** of all strategy-based training samples. We retrained the model on this reduced dataset and evaluated its safety capabilities using jailbreak prompts generated from these strategies. The comparison results are presented in **Rebuttal Table 3**.
> In the table, **ARMOR (seen strategy)** refers to the model trained with access to these strategies, **ARMOR (unseen strategy)** refers to the model trained without exposure to them, and **ARMOR (update strategy)** denotes the model that is provided with these strategies through the strategy library at inference time. As shown in the results, even without seeing these strategies during training, ARMOR still achieves better safety performance than STAR and STAIR. There are two main reasons for this.
> First, although the strategy library enhances ARMOR's accuracy in extracting the true intent, ARMOR's safety performance does not rely solely on it. ARMOR's core remains **intent analysis**, and the strategy library ultimately serves this purpose, which is acting as background knowledge that assists in extracting the true intent.
> Second, ARMOR has learned to identify the essence of jailbreak prompts, which is attempting to hide harmful intent through various forms of obfuscation and this essence is shared across any kind of unseen jailbreak strategies. Therefore, ARMOR can still make comparatively accurate intent inferences even when confronted with jailbreak strategies it has never seen before. In other words, ARMOR's robustness comes from its ability to detect the hidden intent within the prompt itself, rather than from memorizing specific jailbreak strategies.
> Naturally, a larger strategy library improves ARMOR's ability to perform intent extraction across diverse jailbreak prompts. Notably, the results show that by adding the excluded strategies to ARMOR's strategy library at inference time, ARMOR can regain even higher levels of safety, demonstrating its strong capability to effectively utilize the strategy library.
> **Rebuttal Table 3.** Unseen jailbreak strategy
> |Strategy|Qwen-2.5|star|stair| armor (seen strategy)|armor (unseen strategy)| armor (update strategy)|
> |--|--|--|--|--|--|--|
> | Coded Language Obfuscation| 0.38 |0.08 |0.18 |0.0 | 0.04 |0.0||
> | Role-Based Compliance Priming| 0.29 |  0.09| 0.12 | 0.0 | 0.02| 0.01|
> |Format-Based Obfuscation |0.44 | 0.14 | 0.11 | 0.0 | 0.04 | 0.0|
>
> Additionally, we also hope to highlight that the main evaluation in terms of unseen strategies we conducted in our paper is the optimization-based attacks (e.g., the adversarial reasoning and AutoDAN-turbo). These methods iteratively and dynamically update jailbreak prompts through interaction with the victim LLM. For example, AutoDAN-Turbo continuously generates new jailbreak prompts and identifies emerging jailbreak strategies. This iterative process enables these attacks to produce jailbreak prompts under previously unseen strategies. Therefore, ARMOR's robustness against these optimization-based attacks demonstrates its generalizability to unseen jailbreak strategies while other baseline methods show poor safety results against these optimization-based attacks (**Rebuttal Table 4** and **Table 1** of the main paper).
>
> **Rebuttal Table 4.** Jailbreak Attack
> |Jail Method|Dataset|Qwen-2.5 | stair | armor | armor-think|
> |--|--|--|--|--|--|
> |AutoDAN-T|Advbench| 0.96 | 0.25 |  0.05 |  0.03|
> |   |HarmQA| 0.93 | 0.17 |  0.02 |0.02|
> |AdvReason|Advbench| 0.95|  0.48|   0.06| 0.07|
> |   |HarmQA| 0.95|  0.31|   0.00 |  0.00|

---

> ### Author Response · Authors · 2025-11-27
>
> #### **Weakness 4. Multi-turn Jailbreak Attacks.**
> Thank you for the suggestion, this is an interesting question. We evaluated ARMOR's robustness against multi-turn jailbreak attacks using a classic method from the Multi-turn Attack setting, Chain-of-Attack (CoA), on AdvBench. The results, shown in **Rebuttal Table 5**, indicate that CoA-style multi-turn attacks are relatively ineffective against ARMOR, with ASR of 0.02. This is because multi-turn attacks essentially treat the full conversation which comprising both model and user turns, as a single jailbreak prompt. Under this formulation, ARMOR can easily detect the malicious intent embedded in the context and respond with a harmless output.
>
>
> **Rebuttal Table 5.** Multi-turn Attack
> |Model|Qwen-2.5 | star | stair | armor | armor-think|
> |--|--|--|--|--|--|
> |CoA ASR| 0.55  | 0.08| 0.06|0.02|  0.0|
>
>
> #### **Weakness 5. Evaluation on Difference Model Scale.**
>
>
> We further extend the evaluation of the 14B version of ARMOR to additional safety benchmarks, optimization-based jailbreak attacks, and utility assessments. The results, presented in **Rebuttal Table 6**, show that ARMOR built upon Qwen-2.5-14B continues to achieve strong safety performance across all benchmarks and jailbreak attacks. Moreover, it outperforms the 7B model in jailbreak resistance (0.02 and 0.02 vs. 0.08 and 0.04). The enhanced semantic understanding of the 14B model enables ARMOR to better infer user intent and reasoning strategies, thereby strengthening its safety detection capabilities.
>
>
> Similarly, due to the presence of safety tax, ARMOR experiences a slight performance drop in utility compared to its base model, but this reduction is minimal and does not affect overall usability.
>
>
> **Rebuttal Table 6.** 14B scale model
> |Benchmark|    Qwen-2.5-14B    | + ARMOR |
> |--|--|--|
> |AdvReason| 0.84 | 0.020    |
> |AutoDAN-Turbo| 0.920   |0.020|
> |PAIR| 0.106    |   0.028|
> |Wildjailbreak| 0.643|0.006 |
> |JailbreakV|    0.495|0.006 |
> |HarmfulQA|0.024    |0.000  |
> |Beaver|0.022   |0.007  |
> |HarmfulQA|0.024    |0.000  |
>
>
> |Benchmark|    Qwen-2.5-14B    | + ARMOR |
> |--|--|--|
> |MATH| 0.84 | 0.81 |
> |GSM8k| 0.95 | 0.89 |
> |Alpaca|0.96 | 0.94|
> |TruthfulQA| 0.63|0.58 |

---

### Author Response · Authors · 2025-12-03

We sincerely thank all reviewers for their constructive and thoughtful feedback. Before the rebuttal, the scores were: PjoN (6), NjU7 (6), 7bDd (4), and 7bBM (6).

After the rebuttal, Reviewer NjU7 and Reviewer 7bBM explicitly confirmed that our responses fully addressed their concerns. In addition, NjU7 increased the confidence from 2 to 3, and 7bBM increased the score from 6 to 8, further validating the effectiveness of our rebuttal. For the reviewers who have not yet responded, we summarize their concerns and our rebuttal as follows.
### **Response to Reviewer 7bDd**
Reviewer 7bDd gave us 4 but did not respond during the rebuttal. We summarize concerns of Reviewer 7bDd and our responses as follows and we believe that these concerns have been fully addressed.
#### **necessity of external Strategy Library & Generalization to Unseen Jailbreak Strategies**
In the rebuttal, we show that ARMOR's safety strength does not primarily come from the strategy library but the intent analysis and reasoning process. We show that ARMOR's reasoning module remains effective and still outperforms baseline methods even without a strategy library (table 3 in the paper) .
We also show that even without seeing these strategies during training, and even attack strategies are not included in the strategies library, ARMOR still achieves better safety performance than baselines (in table 6 and table 14 in the appendix) for these unseen attacks. We provided detailed reasons about them in our rebuttal.
We also emphasized ARMOR's ability to update the strategy library at any time is a major advantage. When new jailbreak strategies emerge, updating the strategy library within the system prompt is vastly simpler than repeatedly performing model-level safety alignment through retraining, providing an approach far more efficient than retraining the model.
Notably, **Reviewer NjU7 raised the same concern, and explicitly confirmed that our rebuttal sufficiently addressed it**.


#### **More Adaptive Jailbreak Data & Multi-turn Attacks**
The second concern is for more evaluation on adaptive and multi-turn attacks.
We added experiments including (1) more adaptive jailbreak results across multiple datasets, and (2) multi-turn attack results. The new experiments align with the main claims of the paper and demonstrate consistent robustness.


#### **Evaluation Metrics**
The third concern is about the LLM-as-a-Judge used for evaluation in the experiments. We show that the metric used in our experiments is consistent to previous works, and we conducted a human agreement study on the evaluation of LLM-as-a-Judge. **Reviewer 7bBM had an identical concern and explicitly confirmed that our rebuttal fully addressed it**.




### **Response to Reviewer PjoN**
Reviewer PjoN gave us 6 but did not respond during the rebuttal. For each concern of Reviewer PjoN, we provided targeted additional experiments, including:


- Inference-time overhead
- More utility benchmarks
- Unseen strategies
- Multi-turn jailbreak attack
- Different model scale


These additional experiments consistently support the claims of the paper. We believe that these concerns have been fully addressed.




#### **Summary**
Across all reviewers, we believe our rebuttal has fully addressed the concerns. Two reviewers (NjU7 and 7bBM) explicitly confirmed that our responses sufficiently addressed their issues, with one increasing confidence and the other increasing their score.
Last, we summarize the novelty and impact of our paper, which are also emphasized by multiple reviewers. Firstly, ARMOR introduces an intent-centric defense paradigm that identifies and reasons about the core malicious intent behind jailbreaks. Secondly, the external, updatable strategy library enables scalable and interpretable safety improvements without retraining. Finally, ARMOR achieves state-of-the-art safety with low ASR while preserving utility, and generalizes to adaptive and unseen attacks.

---

### Meta-Review · Area_Chair_8G7t · 2026-01-09

**Summary:**

Reviewers broadly agree that this paper makes a strong and timely contribution to LLM safety alignment by introducing an inference-time defense framework against advanced and out-of-distribution jailbreak attacks. The core idea (i.e., explicitly extracting and reasoning over malicious intent via a structured multi-step pipeline) is viewed as both conceptually sound and practically impactful. Across reviews, the empirical results demonstrating substantially reduced attack success rates against adaptive, optimization-based jailbreaks were particularly appreciated. The proposed extensions, including an updatable strategy library and the ARMOR-Think variant, further strengthen the work’s practicality and scalability. Overall, the reviewers view the paper as a meaningful advance in practical LLM safety.

**Reviewer Concerns:**

Several reviewers initially raised concerns regarding reliance on the external strategy library, generalization to unseen jailbreak strategies, evaluation breadth (e.g., multi-turn and adaptive attacks), inference-time overhead, utility degradation, and the use of LLM-as-a-judge without human validation. The rebuttal substantively addressed these issues through additional ablations, expanded experiments, and clarifications. In particular, the authors demonstrated that ARMOR’s robustness does not depend solely on the strategy library, added evaluations on unseen strategies, multi-turn attacks, different model scales, and white-box attacks, and provided quantitative overhead and GPU cost analyses. Human agreement studies were also introduced to validate the evaluation protocol. While some reviewers still note that long-term maintenance of the strategy library and broader utility evaluations remain open research questions, these are generally viewed as limitations of scope rather than flaws undermining the paper’s core contributions.

**Reviewer Scores:**

Two reviewers explicitly confirmed their positive attitude toward this paper during discussion (score: 66->68), leading to final score of 6648. Reviewers that did not yet respond during rebuttal had their major concerns addressed through added experiments and clarifications. Given the strengthened empirical support and response to key weaknesses, no further negative score changes are expected, and the final reviewer scores collectively support acceptance.

---

### Decision · Program_Chairs · 2026-01-26

Accept (Poster)